# Euler Wavelet Method as a Numerical Approach for the Solution of Nonlinear Systems of Fractional Differential Equations

**Sadiye Nergis Tural Polat [1] and Arzu Turan Dincel [2,\*]**

[1]  Department of Electronics and Communications Engineering, Yildiz Technical University, Istanbul 34220, Turkey; nergis@yildiz.edu.tr
[2]  Department of Mathematical Engineering, Yildiz Technical University, Istanbul 34220, Turkey
\*  Correspondence: artur@yildiz.edu.tr

**Abstract:** In this paper, a numerical approach for solving systems of nonlinear fractional differential equations (FDEs) is presented Using the Euler wavelets technique and associated operational matrices for fractional integration, we try to solve those systems of FDEs. The method's major objective is to transform the nonlinear FDE into a nonlinear system of algebraic equations that is straightforward to solve with matrix techniques. The Euler wavelets are constructed using Euler polynomials, which have fewer terms than most other polynomials used to construct other types of wavelets, therefore, using Euler wavelets for the numerical approach provides sparse operational matrices. Thanks to the sparsity of those operational matrices, the proposed numerical approach requires less computation and takes less time to evaluate. The approach described here is also applicable to systems of fractional differential equations with variable orders. To illustrate the strength and performance of the method, four numerical examples are provided.

**Keywords:** systems of fractional differential equations; Euler wavelet; numerical solution; operational matrix method

## 1. Introduction

Fractional calculus has drawn growing attention for decades since it is essential to many fields of research and engineering. Due to the fact that fractional order differential operators are nonlocal operators, compared to integer order differential equations, fractional differential equations have the benefit of being able to characterize some natural physics processes and dynamic system processes better. The use of FDEs allows for the elegant modeling of numerous physics, applied mathematics, and engineering systems [1–15]. However, for most of those cases, analytical solutions to those FDE systems are very hard to obtain. Therefore, there has been a considerable effort to obtain numerical solutions to ordinary and partial FDEs. As a result, a variety of numerical methods have been developed by many researchers recently. Those methods include the spline collocation method [16], B-spline method [17], block-pulse function method [18], Laplace transform method [19], variational iteration method [20], finite difference method [21], Adomian decomposition method [22], Mellin transform method [23], homotopy perturbation method [24], Laplace optimized decomposition method [25], predictor–corrector approach [26], fractional complex transform method [27], computational matrix method [28], spectral collocation methods [29], Legendre spectral-collocation method [30], block-by-block method [31], and finite element method [32].

Most of those methods are applied to equations having only one or a few fractional differential terms in them. Therefore, they become impractical and cumbersome for the multiterm or variable-order FDEs. What is more, the finite elements method approximates the differential equation whereas the finite differences and spectral methods approximate the solution of the differential equation. Spectral base functions are infinitely differentiable, though they are nonzero for all of the regions of interest. On the other hand, the base

functions used in the finite differences and finite elements methods have compact support though they are not necessarily continuous. The wavelet methods have the upper hand compared to those methods such that they have both compact support and they are smooth functions.

The wavelet theory has received a lot of attention in recent years. There are a lot of wavelet methods proposed for numerical approximations such as Haar, Legendre, Chebyshev, Euler, sine-cosine, and Gegenbauer wavelets. With the use of the orthogonal basis of those wavelets, it is possible to reduce the current problem to a set of either linear or nonlinear algebraic equations. Large systems of algebraic equations may require more computational load. The simplest wavelet method is the Haar wavelet and thanks to their simplicity they are commonly preferred. The drawback for the Haar wavelets is their lower accuracy. Sine-cosine wavelets produce better results for periodic functions. Therefore, their applications are limited.

In this paper, we focus on the Euler wavelet approximation to explore the advantages of the method. Since the Euler wavelets are produced using Euler polynomials, and since Euler polynomials generally have fewer terms than other polynomials used for the other wavelet methods, they produce sparser operational matrices for the numerical approach. Sparse operational matrices are essential for fast evaluation and low computational requirements in the numerical approach, especially for the large collocation points used to increase accuracy.

The theory of elasticity, dynamics, fluid mechanics, oscillation, and quantum dynamics are only a few examples of the topics where the system of fractional differential equations is frequently utilized to model the system behavior. In this paper, we aim to numerically solve the system of FDEs in the form of:

$$D_*^{\alpha_i} u_i(x) = f_i(x, u_1, u_2, \ldots, u_n), \ u_i^{(r)}(0) = c_i, \ 1 \leq i \leq n, \ 0 \leq r \leq \lceil \alpha_i \rceil \tag{1}$$

where $m_i - 1 < \alpha_i \leq m_i$, $m_i \in \mathbb{Z}^+$, $D_*^{\alpha_i}$ is the Caputo fractional differential operator.

The paper is organized as follows. The brief definitions and basic properties of fractional calculus are presented in Section 2. In Section 3, Euler wavelets are introduced and the operational matrices for the numerical integration for the solution of (one) are constructed. Several numerical examples are presented in Section 4 to demonstrate the validity of the method. The paper concludes by stating the important results of the method in Section 5.

## 2. Fundamentals of the Fractional Calculus

**Definition 1.** *The Riemann–Liouville fractional integral operator of order $\alpha$ is given as*

$$I^\alpha f(t) = \left\{ \begin{array}{cc} \frac{1}{\Gamma(\alpha)} \int\limits_0^t \frac{f(\tau)}{(t-\tau)^{1-\alpha}} d\tau, & \alpha > 0, \ t > 0 \\ f(t) & , \ \alpha = 0 \end{array} \right\} \tag{2}$$

For $\alpha \geq 0$, $\beta \geq 0$, $a \geq 0$ and $\eta \geq -1$ we have the following properties of the Riemann–Liouville fractional integral

$$\text{(i) } I^\alpha I^\beta = I^\beta I^\alpha \tag{3}$$

$$\text{(ii) } I^\alpha \left( I^\beta f(t) \right) = I^\beta (I^\alpha f(t)) = I^{\alpha+\beta} f(t) \tag{4}$$

$$\text{(iii) } I^\alpha (t-a)^\eta = \frac{\Gamma(\eta+1)}{\Gamma(\alpha+\eta+1)} (t-a)^{\alpha+\eta} \tag{5}$$

The Riemann–Liouville fractional derivative is defined by

$$D^\alpha f(t) = \left(\frac{d}{dt}\right)^n \left(I^{n-\alpha} f(t)\right), \; 0 \leq n-1 < \alpha \leq n \tag{6}$$

where $n$ is an integer and $t > 0$. When representing real-world processes, the derivative of the Riemann–Liouville operator has a few drawbacks. As a result, in this study, we employ Caputo's modified fractional differential operator $D^\alpha$, which is described in the formulation that follows.

**Definition 2.** *The following expression is the fractional derivative operator defined by Caputo:*

$$D^\alpha f(t) = \left\{ \begin{array}{ll} \frac{d^n f(t)}{dt^n} & , \; \alpha = n \in R \\ \frac{1}{\Gamma(n-\alpha)} \int\limits_0^t \frac{f^{(n)}(t)}{(t-\tau)^{1-n+\alpha}} d\tau & , \; 0 \leq n-1 < \alpha \leq n \end{array} \right\} \tag{7}$$

The following two common equations can be used to express the relationship between the Riemann–Liouville operator and the Caputo operator:

$$D^\alpha I^\alpha f(t) = f(t) \tag{8}$$

and

$$I^\alpha D^\alpha f(t) = f(t) - \sum_{k=0}^{n-1} f^{(k)}\left(0^+\right) \frac{t^k}{k!} \tag{9}$$

For more information on fractional differentiation and integration, the reader is directed to [5].

## 3. Euler Wavelets and Derivation of Operational Matrices for Euler Wavelets
### 3.1. Euler Wavelets

Localized wavelike functions known as "wavelets" are used in wavelet analysis. A mother wavelet and its enlarged and translated variations make up a family of wavelets. We may create the following family of continuous wavelets as [33] by continually varying the translation parameter b and the dilation parameter a.

$$\psi_{a,b}(t) = |a|^{-1/2} \psi\left(\frac{t-b}{a}\right), \qquad a, b \in R, \quad a \neq 0 \tag{10}$$

When the parameters for translation and dilation are chosen to have discrete values $a = a_0^{-k}, b = nb_0 a_0^{-k}, a_0 > 1, b_0 > 0$, we obtain the discrete wavelets as

$$\psi_{kn}(t) = |a_0|^{k/2} \psi\left(a_0^k t - nb_0\right) \tag{11}$$

Euler wavelets $\psi_{nm} = \psi(k, \tilde{n}, m, t)$ have four arguments: $\tilde{n} = n-1, n = 1, 2, 3, \cdots, 2^{k-1}$, $m$ is the order for Euler polynomials, $t$ is the normalized time, and $k$ can take any positive integer value. Euler wavelets for $t \in [0, 1)$ results

$$\psi_{nm}(t) = \left\{ \begin{array}{ll} 2^{\frac{k-1}{2}} \widetilde{E}_m\left(2^{k-1}t - n + 1\right), & \frac{n-1}{2^{k-1}} \leq t < \frac{n}{2^{k-1}} \\ 0 & , \; otherwise \end{array} \right\} \tag{12}$$

and

$$\widetilde{E}_m(t) = \left\{ \begin{array}{ll} 1 & , \; m = 0 \\ \frac{1}{\sqrt{\frac{2(-1)^{m-1}(m!)^2}{(2m)!} E_{2m+1}(0)}} & , \; m > 0 \end{array} \right\} \tag{13}$$

where $m = 0, 1, 2, \ldots, M - 1$ and $n = 1, 2, 3, \ldots, 2^{k-1}$ and $E_m(t)$ are the Euler polynomials defined by [34]

$$\sum_{k=0}^{m} \binom{m}{k} E_k(t) + E_m(t) = 2t^m \tag{14}$$

The Euler polynomials start with the polynomials given in (15)

$$E_0(t) = 1, \; E_1(t) = t - \frac{1}{2}, \; E_2(t) = t^2 - t, \; E_3(t) = t^3 - \frac{2}{3}t^2 + \frac{1}{4}, \ldots \tag{15}$$

*3.2. Function Approximation*

Euler wavelets can be used to represent a function $f(t), t \in [0, 1)$ as in the following:

$$f(t) = \sum_{n=1}^{2^{k-1}} \sum_{m=0}^{M-1} c_{nm} \Psi_{nm}(t) = C^T \psi(t) \tag{16}$$

where $T$ represents transposition and

$$C = \left[ c_{10}, \; c_{11}, \; \ldots c_{1(M-1)}, \; c_{20}, \; c_{21}, \; \ldots c_{2(M-1)} \ldots c_{2^{k-1}0}, \; c_{2^{k-1}1}, \; \ldots c_{2^{k-1}(M-1)} \right]^T \tag{17}$$

$$\Psi = \left[ \Psi_{10}, \; \Psi_{11}, \; \ldots \Psi_{1(M-1)}, \; \Psi_{20}, \; \Psi_{21}, \; \ldots \Psi_{2(M-1)} \ldots \Psi_{2^{k-1}0}, \; \Psi_{2^{k-1}1}, \; \ldots \Psi_{2^{k-1}(M-1)} \right]^T \tag{18}$$

Here, we define the total number of discrete collocation points $t_i = \frac{i - 0.5}{m'}$ $i = 1, 2, 3, \ldots,$ $m'$ as $m' = 2^{k-1}M$. Using those points, the Euler wavelet matrix $\phi_{m' x m'}$ becomes

$$\phi_{m' x m'} = \left[ \Psi(t_1) \; \Psi(t_2) \; \Psi(t_3) \; \cdots \; \Psi(t_{m'}) \right] \tag{19}$$

The Euler wavelet matrix for $k = 2$, $M = 3$, and $\alpha = 0.5$ yields

$$\phi_{m' x m'} = \begin{bmatrix} 1.4142 & 1.4142 & 1.4142 & 0 & 0 & 0 \\ -0.9428 & 0 & 0.9428 & 0 & 0 & 0 \\ -0.4811 & -0.8660 & -0.4811 & 0 & 0 & 0 \\ 0 & 0 & 0 & 1.4142 & 1.4142 & 1.4142 \\ 0 & 0 & 0 & -0.9428 & 0 & 0.9428 \\ 0 & 0 & 0 & -0.4811 & -0.8660 & -0.4811 \end{bmatrix} \tag{20}$$

*3.3. Euler Wavelet Operational Matrix of Fractional Integration*
Block Pulse Functions

The block pulse functions (BPFs) of set $m'$ are given by

$$b_i(t) = \left\{ \begin{array}{ll} 1, & \frac{i-1}{m'} \leq t < \frac{i}{m'} \\ 0, & otherwise \end{array} \right\} \tag{21}$$

where $i = 1, 2, 3, \ldots, m'$. The functions $b_i(t)$ are both orthogonal and disjoint. For $t \in [0, 1)$

$$b_i(t)b_j(t) = \left\{ \begin{array}{ll} 0, & i \neq j \\ b_i(t), & i = j \end{array} \right\} \tag{22}$$

$$\int_0^1 b_i(\tau)b_j(\tau)d\tau = \left\{ \begin{array}{ll} 0, & i \neq j \\ 1/m', & i = j \end{array} \right\} \tag{23}$$

We can use the $m'$ set of BPFs to represent a function $f(t), t \in [0, 1)$ provided that $f(t)$ is square integrable in the represented interval such that

$$f(t) = \sum_{i=1}^{m'} f_i \, b_i(t) = f^T B_{m'}(t) \tag{24}$$

where $f = [f_1, \, f_2, \, \ldots, f_{m'}]^T$, $B_{m'}(t) = [b_1(t), \, b_2(t), \, \ldots, b_{m'}(t)]^T$ and $f_i$ are

$$f_i = \frac{1}{m'} \int_{(i-1)/m'}^{i/m'} f(t) \, b_i(t) dt \tag{25}$$

Similarly, the Euler wavelet matrix can also be represented with an $m'$ set of BPFs as

$$\psi(t) = \phi_{m'xm'} \, B_{m'}(t) \tag{26}$$

Now, we use the definition of the block pulse operational matrix for fractional integration $F^\alpha$ given in [35]

$$I^\alpha B_{m'}(t) \approx F^\alpha \, B_{m'}(t) \tag{27}$$

where

$$F^\alpha = \frac{1}{m^\alpha} \frac{1}{\Gamma(\alpha+2)} \begin{bmatrix} 1 & \xi_1 & \xi_2 & \xi_3 \cdots \xi_{m'-1} \\ 0 & 1 & \xi_1 & \xi_2 \cdots \xi_{m'-2} \\ 0 & 0 & 1 & \xi_1 \cdots \xi_{m'-3} \\ \vdots & \vdots & \ddots \ddots \vdots & \vdots \\ 0 & 0 & \cdots 0 & 1 & \xi_1 \\ 0 & 0 & \cdots 0 & 0 & 1 \end{bmatrix} \tag{28}$$

with $\xi_k = (k+1)^{\alpha+1} - 2k^{\alpha+1} + (k-1)^{\alpha+1}$. Using (26)–(27) we can write

$$I^\alpha \psi(t) \approx I^\alpha \phi_{m'xm'} \, B_{m'}(t) = \phi_{m'xm'} \, I^\alpha B_{m'}(t) \approx \phi_{m'xm'} F^\alpha B_{m'}(t) \approx \phi_{m'xm'} F^\alpha \phi_{m'xm'}^{-1} \psi(t) \tag{29}$$

$$I^\alpha \psi(t) \approx P_{m' \times m'}^\alpha \, \psi(t) \tag{30}$$

$$P_{m'xm'}^\alpha \approx \phi_{m'xm'} F^\alpha \phi_{m'xm'}^{-1} \tag{31}$$

where matrix $P_{m' \times m'}^\alpha$ is called the Euler wavelet operational matrix of fractional integration. As an example, the operational matrix for $\alpha = 0.5, k = 2, M = 3$ yields:

$$P_{m' \times m'}^\alpha = \begin{bmatrix} 0.4616 & 0.3150 & -0.1631 & 0.5012 & -0.1509 & 0.1404 \\ 0.0878 & 0.2243 & 0.4203 & 0.0717 & -0.0449 & 0.0626 \\ -0.1305 & -0.1591 & 0.2354 & -0.2110 & 0.0615 & -0.0545 \\ 0 & 0 & 0 & 0.4616 & 0.3150 & -0.1631 \\ 0 & 0 & 0 & 0.0878 & 0.2243 & 0.4203 \\ 0 & 0 & 0 & -0.1305 & -0.1591 & 0.2354 \end{bmatrix} \tag{32}$$

Let us point out that, considering the two vectors given as $P = [p_1, p_2, \ldots p_{m'}]^T$, $R = [r_1, r_2, \ldots r_{m'}]^T$ we have $P * R = [p_1 r_1, p_2 r_2, \ldots p_{m'} r_{m'}]^T$ and $P^n = [p_1^n, p_2^n, \ldots p_{m'}^n]^T$ for the multiplication and the $n$th power of those vectors thanks to the properties of the BPFs.
The convergence analysis of the Euler wavelets can be found in [33].

## 4. Numerical Examples

Here, four nonlinear systems of FDEs are solved using the proposed method to point out the accuracy and efficiency of the method.

*4.1. Example 1*

First, consider the system of FDE defined as:

$$
\begin{aligned}
D^\alpha u(t) &= \tfrac{3}{4} v^2(t) & , \; 0 < \alpha \le 1 \\
D^\beta v(t) &= u(t)v(t) - \tfrac{v^4(t)}{8} + 2, \; 0 < \beta \le 1
\end{aligned}
\tag{33}
$$

With the initial values $u(0) = 0$, $v(0) = 0$ and the exact solution for $\alpha = 1$ and $\beta = 1$ is given as $u_{ex}(t) = t^3$, $v_{ex}(t) = 2t$, $t \in [0,1]$.

Now, applying the proposed method to the fractional derivatives, we have

$$
\begin{aligned}
D^\alpha u(t) &\approx R_{m'}^T \, \psi(t) \\
D^\beta v(t) &\approx S_{m'}^T \, \psi(t)
\end{aligned}
\tag{34}
$$

where $R_{m'}^T = [r_1, r_2, \ldots, r_{m'}]$ and $S_{m'}^T = [s_1, s_2, \ldots, s_{m'}]$ are the unknown coefficients. Using (9), (26), (30), (34), and the initial conditions we obtain

$$
u(t) = I^\alpha D^\alpha u(t) + u(0) \approx R_{m'}^T \, P_{m' \times m'}^\alpha \psi(t) \approx \underbrace{R_{m'}^T \, P_{m' \times m'}^\alpha \phi_{m' \times m'}}_{H_{m'}^T} B_{m'}(t)
$$

$$
v(t) = I^\beta D^\beta v(t) + v(0) \approx S_{m'}^T \, P_{m' \times m'}^\beta \psi(t) \approx \underbrace{S_{m'}^T \, P_{m' \times m'}^\beta \phi_{m' \times m'}}_{K_{m'}^T} B_{m'}(t)
\tag{35}
$$

where $H_{m'}^T = [h_1, h_2, \ldots, h_{m'}]$ $K_{m'}^T = [k_1, k_2, \ldots, k_{m'}]$ are also the vectors of size $1 \times m'$

$$
\begin{aligned}
v^4(t) &\approx \left(K_{m'}^T\right)^4 B_{m'}(t) \\
u(t)v(t) &\approx \left(H_{m'}^T * K_{m'}^T\right) B_{m'}(t)
\end{aligned}
\tag{36}
$$

Finally, using (34)–(36) in (33) we obtain the system of algebraic equations with $2m'$ unknowns in the form of $R_{m'}^T$ and $S_{m'}^T$ coefficients:

$$
\begin{aligned}
R_{m'}^T \, \phi_{m' \times m'} &= \tfrac{3}{4} \left(K_{m'}^T\right)^2, \\
S_{m'}^T \, \phi_{m' \times m'} &= H_{m'}^T * K_{m'}^T - \tfrac{1}{8} \left(K_{m'}^T\right)^4 + [2, 2, \ldots, 2]_{1 \times m'}
\end{aligned}
\tag{37}
$$

Solving (37) for the $R_{m'}^T$ and $S_{m'}^T$ coefficients also provides the numerical solution for $u(t)$ and $v(t)$, as indicated in (35).

Table 1 summarizes the absolute errors obtained from the proposed method for $u(t)$ and $v(t)$ for several $m'$ values ($\alpha = 1$, $\beta = 1$). $E_u$ and $E_v$ represent the absolute errors in $u(t)$ and $v(t)$, respectively. We present the solution graphs $u(t)$ and $v(t)$ for integer orders with the exact solution in Figure 1 and the fractional orders $\alpha = 0.7, 0.8, 0.9$, $\beta = 0.7, 0.8, 0.9$ with the integer orders $\alpha = 1$, $\beta = 1$ in Figure 2. As Table 1 and Figure 1 suggest, the proposed method follows the exact solution closely for the integer-orders of $\alpha = 1$, $\beta = 1$. Also, as can be seen from the table, the absolute errors decrease with the larger $m'$ values, as expected. The absolute errors are approximately on the order of $10^{-4}$ for $m' = 24$, on the order of $10^{-5}$ for $m' = 48$, and on the order of $10^{-5}$–$10^{-6}$ for $m' = 96$. In Figure 2, the fractional orders $\alpha, \beta$ are chosen to be equal, though they can be chosen arbitrarily on the interval $(0, 1]$. Several different $\alpha, \beta$ values are chosen in the following examples. As Figure 2 demonstrates, the solution approaches to the exact solution when $\alpha, \beta$ approach one. As a result, Table 1 and Figures 1 and 2 suggest that the proposed method is a valid approximation to the system of FDE in question.

**Table 1.** Absolute errors $E_u$ and $E_v$ corresponding to $u(t)$ and $v(t)$ for several $m'$ values of example 1.

| $t$ | $m'=12$ | | $m'=24$ | | $m'=48$ | | $m'=96$ | |
|---|---|---|---|---|---|---|---|---|
| | $E_u$ | $E_v$ | $E_u$ | $E_v$ | $E_u$ | $E_v$ | $E_u$ | $E_v$ |
| 0 | $1.09 \times 10^{-3}$ | $2.51 \times 10^{-6}$ | $1.36 \times 10^{-4}$ | $7.85 \times 10^{-8}$ | $1.70 \times 10^{-5}$ | $2.45 \times 10^{-9}$ | $2.12 \times 10^{-6}$ | $7.67 \times 10^{-11}$ |
| 0.1 | $1.89 \times 10^{-4}$ | $2.75 \times 10^{-6}$ | $9.92 \times 10^{-5}$ | $6.36 \times 10^{-7}$ | $2.42 \times 10^{-5}$ | $1.48 \times 10^{-7}$ | $5.23 \times 10^{-6}$ | $3.64 \times 10^{-8}$ |
| 0.2 | $7.94 \times 10^{-4}$ | $2.03 \times 10^{-5}$ | $1.93 \times 10^{-4}$ | $4.74 \times 10^{-6}$ | $4.19 \times 10^{-5}$ | $1.16 \times 10^{-6}$ | $1.05 \times 10^{-5}$ | $2.90 \times 10^{-7}$ |
| 0.3 | $9.47 \times 10^{-4}$ | $6.46 \times 10^{-5}$ | $2.42 \times 10^{-4}$ | $1.57 \times 10^{-5}$ | $6.69 \times 10^{-5}$ | $3.91 \times 10^{-6}$ | $1.66 \times 10^{-5}$ | $9.75 \times 10^{-7}$ |
| 0.4 | $1.56 \times 10^{-3}$ | $1.51 \times 10^{-4}$ | $3.38 \times 10^{-4}$ | $3.70 \times 10^{-5}$ | $8.52 \times 10^{-5}$ | $9.21 \times 10^{-6}$ | $2.21 \times 10^{-5}$ | $2.30 \times 10^{-6}$ |
| 0.5 | $2.87 \times 10^{-3}$ | $2.89 \times 10^{-4}$ | $5.81 \times 10^{-4}$ | $7.14 \times 10^{-5}$ | $1.28 \times 10^{-4}$ | $1.78 \times 10^{-5}$ | $2.99 \times 10^{-5}$ | $4.45 \times 10^{-6}$ |
| 0.6 | $2.04 \times 10^{-3}$ | $4.88 \times 10^{-4}$ | $5.60 \times 10^{-4}$ | $1.21 \times 10^{-4}$ | $1.39 \times 10^{-4}$ | $3.03 \times 10^{-5}$ | $3.40 \times 10^{-5}$ | $7.56 \times 10^{-6}$ |
| 0.7 | $2.76 \times 10^{-3}$ | $7.52 \times 10^{-4}$ | $6.84 \times 10^{-4}$ | $1.87 \times 10^{-4}$ | $1.64 \times 10^{-4}$ | $4.67 \times 10^{-5}$ | $4.12 \times 10^{-5}$ | $1.17 \times 10^{-5}$ |
| 0.8 | $3.12 \times 10^{-3}$ | $1.08 \times 10^{-3}$ | $7.83 \times 10^{-4}$ | $2.68 \times 10^{-4}$ | $2.02 \times 10^{-4}$ | $6.69 \times 10^{-5}$ | $5.04 \times 10^{-5}$ | $1.67 \times 10^{-5}$ |
| 0.9 | $4.05 \times 10^{-3}$ | $1.44 \times 10^{-3}$ | $9.57 \times 10^{-4}$ | $3.60 \times 10^{-4}$ | $2.40 \times 10^{-4}$ | $8.99 \times 10^{-5}$ | $6.07 \times 10^{-5}$ | $2.25 \times 10^{-5}$ |

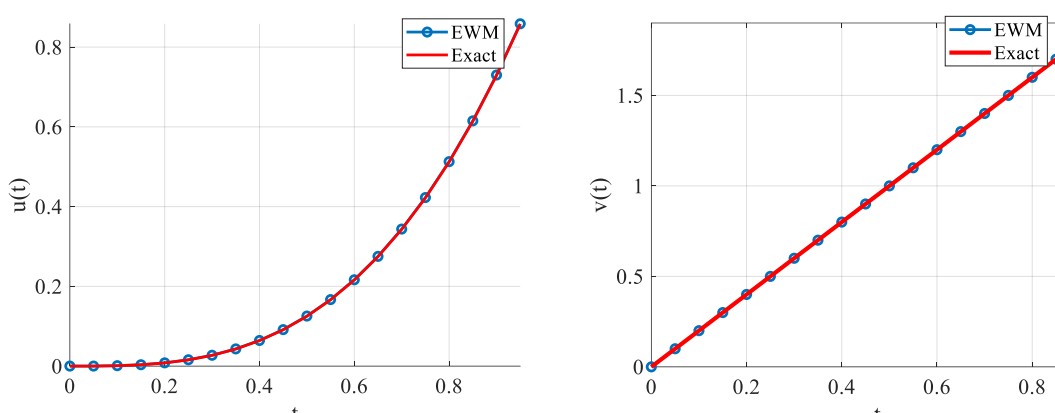

**Figure 1.** Numerical and exact solutions of $u(t)$ and $v(t)$ for $\alpha = 1$, $\beta = 1$ for example, 1 ($m' = 12$).

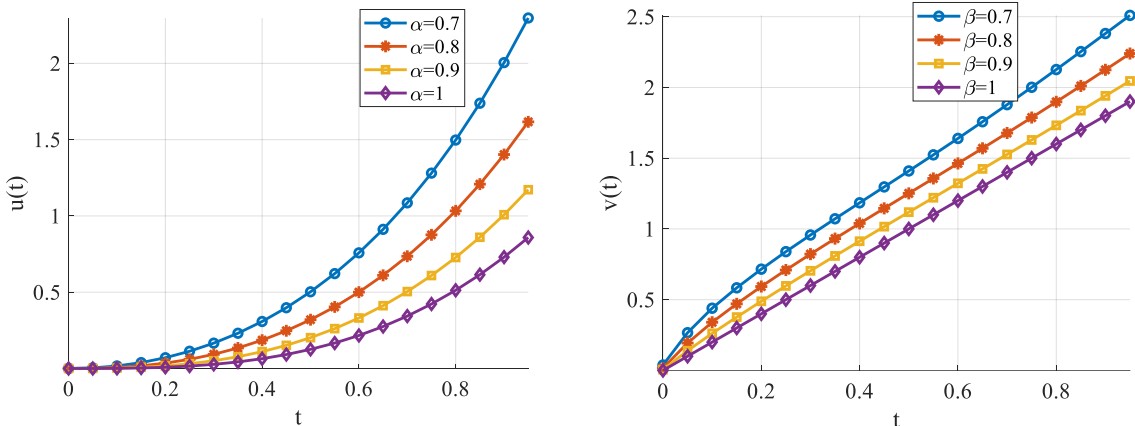

**Figure 2.** Numerical solutions of $u(t)$ and $v(t)$ for several fractional orders for example 1 ($m' = 24$).

*4.2. Example 2*

For the second example, consider the system of FDEs given as:

$$D^\alpha u(t) = -1002u(t) + 1000v^2(t), \ 0 < \alpha \le 1$$
$$D^\beta v(t) = u(t) - v(t) - v^2(t) \quad , \ 0 < \beta \le 1 \tag{38}$$

where the initial values are given as $u(0) = 1$, $v(0) = 1$ and the exact solution for $\alpha = 1$, $\beta = 1$ becomes $u_{ex}(t) = e^{-2t}$, $v_{ex}(t) = e^{-t}$, $t \in [0, 1]$.

As in Example 1, applying the proposed method to the fractional derivatives using (9), (26), (30), (34) and initial conditions for this system of FDEs yield

$$u(t) = I^\alpha D^\alpha u(t) + u(0) \approx R_{m'}^T P_{m' \times m'}^\alpha \psi(t) + 1 \approx \underbrace{R_{m'}^T P_{m' \times m'}^\alpha \phi_{m' \times m'}}_{H_{m'}^T} B_{m'}(t) + 1$$

$$v(t) = I^\beta D^\beta v(t) + v(0) \approx S_{m'}^T P_{m' \times m'}^\beta \psi(t) + 1 \approx \underbrace{S_{m'}^T P_{m' \times m'}^\beta \phi_{m' \times m'}}_{K_{m'}^T} B_{m'}(t) + 1 \qquad (39)$$

and

$$v^2(t) \approx \left(K_{m'}^T + 1\right)^2 B_{m'}(t) \approx \left[\left(K_{m'}^T\right)^2 + 2K_{m'}^T + 1\right] B_{m'}(t) \qquad (40)$$

Finally, substituting (34), (39), and (40) into (38) we obtain the system of algebraic equations with $2m'$ unknowns in the form of $R_{m'}^T$ and $S_{m'}^T$ coefficients:

$$R_{m'}^T \phi_{m' \times m'} = -1002\left(H_{m'}^T + [1, 1, \ldots, 1]_{1 \times m'}\right) + 1000\left[\left(K_{m'}^T\right)^2 + 2K_{m'}^T + [1, 1, \ldots, 1]_{1 \times m'}\right],$$
$$S_{m'}^T \phi_{m' \times m'} = \left(H_{m'}^T + [1, 1, \ldots, 1]_{1 \times m'}\right) - \left(K_{m'}^T + [1, 1, \ldots, 1]_{1 \times m'}\right) - \left[\left(K_{m'}^T\right)^2 + 2K_{m'}^T + [1, 1, \ldots, 1]_{1 \times m'}\right] \qquad (41)$$

$$R_{m'}^T \phi_{m' \times m'} = -1002 H_{m'}^T + 1000\left(K_{m'}^T\right)^2 + 2000 K_{m'}^T - [2, 2, \ldots, 2]_{1 \times m'},$$
$$S_{m'}^T \phi_{m' \times m'} = H_{m'}^T - \left(K_{m'}^T\right)^2 - 3K_{m'}^T - [1, 1, \ldots, 1]_{1 \times m'} \qquad (42)$$

Solving (42) for the $R_{m'}^T$ and $S_{m'}^T$ coefficients also provides the numerical solution for $u(t)$ and $v(t)$, as indicated in (35).

Similar to the first example results, Table 2 summarizes the absolute errors obtained from the proposed method for $u(t)$ and $v(t)$ for several $m'$ values ($\alpha = 1$, $\beta = 1$). $E_u$ and $E_v$ represent the absolute errors in $u(t)$ and $v(t)$, respectively. As can be seen from the table, the absolute errors decrease with the larger $m'$ values, as expected. The exact and proposed method solutions for $\alpha = 1$, $\beta = 1$ are also plotted in Figure 3. The absolute errors are roughly on the order of $10^{-4}$–$10^{-5}$ for $m' = 24$, on the order of $10^{-5}$ for $m' = 48$, and on the order of $10^{-6}$ for $m' = 96$. Table 2 demonstrates that the proposed method follows the exact solution closely for the integer-orders of $\alpha = 1$ and $\beta = 1$. The solution graphs $u(t)$ and $v(t)$ for several fractional orders $\alpha, \beta$ with integer orders $\alpha = 1$, $\beta = 1$ are given in Figure 4. Here, the fractional orders are chosen to be not equal. Numerical simulation is done for the fractional order pairs $\alpha = 0.4$, $\beta = 0.5$, $\alpha = 0.6$, $\beta = 0.7$, $\alpha = 0.8$, $\beta = 0.9$ and integer orders. As Figure 4 demonstrates, the solution approaches to the exact solution when $\alpha, \beta$ approach one, which thus validates the fractional solutions.

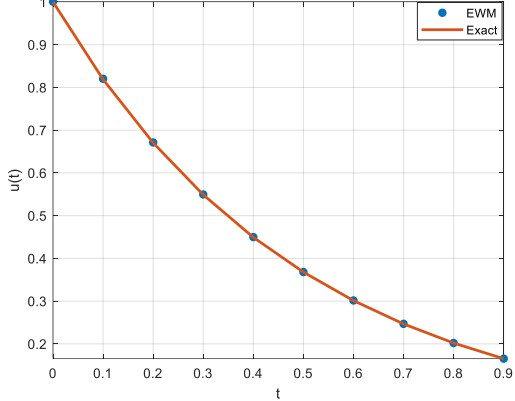
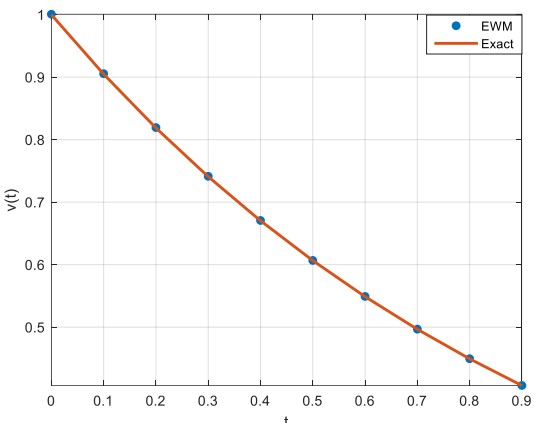

**Figure 3.** Numerical and exact solutions of $u(t)$ and $v(t)$ for $\alpha = 1$, $\beta = 1$ for Example 2 ($m' = 12$).

**Table 2.** Absolute errors $E_u$ and $E_v$ corresponding to $u(t)$ and $v(t)$ for several $m'$ values of Example 2.

| $t$ | $m'=12$ | | $m'=24$ | | $m'=48$ | | $m'=96$ | |
|---|---|---|---|---|---|---|---|---|
| | $E_u$ | $E_v$ | $E_u$ | $E_v$ | $E_u$ | $E_v$ | $E_u$ | $E_v$ |
| 0 | $6.71 \times 10^{-4}$ | $7.06 \times 10^{-4}$ | $3.35 \times 10^{-4}$ | $1.96 \times 10^{-4}$ | $1.17 \times 10^{-4}$ | $5.16 \times 10^{-5}$ | $3.70 \times 10^{-5}$ | $1.32 \times 10^{-5}$ |
| 0.1 | $1.46 \times 10^{-3}$ | $7.59 \times 10^{-4}$ | $3.27 \times 10^{-4}$ | $1.82 \times 10^{-4}$ | $8.40 \times 10^{-5}$ | $4.55 \times 10^{-5}$ | $2.08 \times 10^{-5}$ | $1.15 \times 10^{-5}$ |
| 0.2 | $9.32 \times 10^{-4}$ | $6.03 \times 10^{-4}$ | $2.45 \times 10^{-4}$ | $1.52 \times 10^{-4}$ | $6.32 \times 10^{-5}$ | $3.88 \times 10^{-5}$ | $1.60 \times 10^{-5}$ | $9.68 \times 10^{-6}$ |
| 0.3 | $8.05 \times 10^{-4}$ | $5.27 \times 10^{-4}$ | $1.97 \times 10^{-4}$ | $1.31 \times 10^{-4}$ | $4.68 \times 10^{-5}$ | $3.20 \times 10^{-5}$ | $1.17 \times 10^{-5}$ | $8.01 \times 10^{-6}$ |
| 0.4 | $5.01 \times 10^{-4}$ | $4.10 \times 10^{-4}$ | $1.44 \times 10^{-4}$ | $1.08 \times 10^{-4}$ | $3.70 \times 10^{-5}$ | $2.70 \times 10^{-5}$ | $8.81 \times 10^{-6}$ | $6.65 \times 10^{-6}$ |
| 0.5 | $8.53 \times 10^{-5}$ | $2.51 \times 10^{-4}$ | $6.64 \times 10^{-5}$ | $7.47 \times 10^{-5}$ | $1.89 \times 10^{-5}$ | $2.03 \times 10^{-5}$ | $5.62 \times 10^{-6}$ | $5.28 \times 10^{-6}$ |
| 0.6 | $3.51 \times 10^{-4}$ | $3.00 \times 10^{-4}$ | $7.60 \times 10^{-5}$ | $7.03 \times 10^{-5}$ | $1.86 \times 10^{-5}$ | $1.76 \times 10^{-5}$ | $4.96 \times 10^{-6}$ | $4.49 \times 10^{-6}$ |
| 0.7 | $2.08 \times 10^{-4}$ | $2.21 \times 10^{-4}$ | $5.39 \times 10^{-5}$ | $5.58 \times 10^{-5}$ | $1.47 \times 10^{-5}$ | $1.45 \times 10^{-5}$ | $3.66 \times 10^{-6}$ | $3.62 \times 10^{-6}$ |
| 0.8 | $1.70 \times 10^{-4}$ | $1.89 \times 10^{-4}$ | $4.34 \times 10^{-5}$ | $4.70 \times 10^{-5}$ | $9.75 \times 10^{-6}$ | $1.13 \times 10^{-5}$ | $2.46 \times 10^{-6}$ | $2.82 \times 10^{-6}$ |
| 0.9 | $9.89 \times 10^{-5}$ | $1.30 \times 10^{-4}$ | $2.94 \times 10^{-5}$ | $3.61 \times 10^{-5}$ | $7.67 \times 10^{-6}$ | $8.98 \times 10^{-6}$ | $1.74 \times 10^{-6}$ | $2.19 \times 10^{-6}$ |

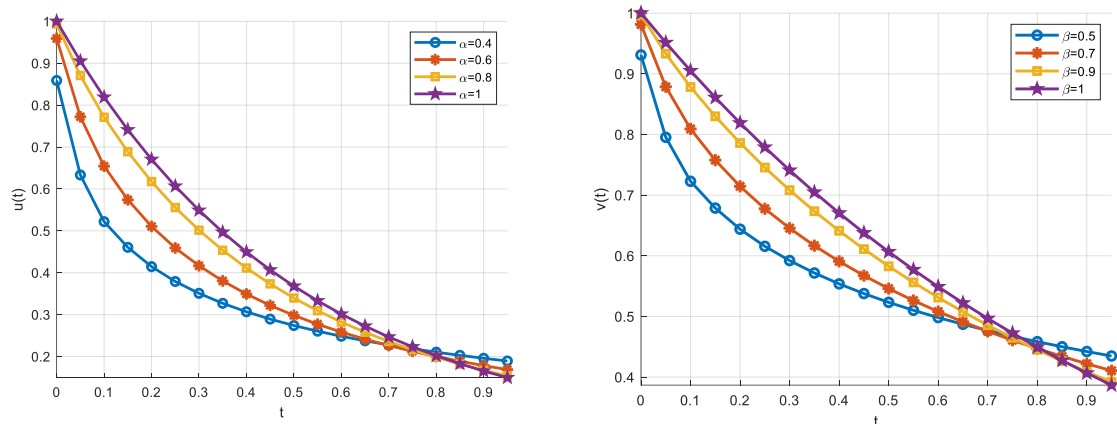

**Figure 4.** Numerical solutions of $u(t)$ and $v(t)$ for several fractional orders for Example 2 ($m' = 24$).

*4.3. Example 3*

Here, we consider the following system of FDEs

$$\begin{array}{ll} D^\alpha u(t) = \dfrac{u(t)}{2} & , \ 0 < \alpha \leq 1 \\ D^\beta v(t) = u^2(t) + v(t), & 0 < \beta \leq 1 \end{array} \tag{43}$$

with $u(0) = 1$, $v(0) = 0$. The exact solution for the integer orders $\alpha = 1$, $\beta = 1$ is given as $u_{ex}(t) = e^{t/2}$, $v_{ex}(t) = te^t$, $t \in [0, 1]$.

Applying the proposed method to this system of FDEs by following the same steps of the previous two examples results in the following system of algebraic equations (44), which is then solved for the unknown coefficients $R_{m'}^T$ and $S_{m'}^T$.

$$\begin{aligned} R_{m'}^T \, \phi_{m' \times m'} &= \tfrac{1}{2}\left(H_{m'}^T + [1, 1, \ldots, 1]_{1 \times m'}\right), \\ S_{m'}^T \, \phi_{m' \times m'} &= \left(H_{m'}^T + [1, 1, \ldots, 1]_{1 \times m'}\right)^2 - \left(K_{m'}^T\right) \\ &= \left(H_{m'}^T\right)^2 + 2H_{m'}^T + [1, 1, \ldots, 1]_{1 \times m'} - K_{m'}^T \end{aligned} \tag{44}$$

The results obtained for $u(t)$ and $v(t)$ for several $m'$ values ($\alpha = 1$, $\beta = 1$) are summarized in Table 3. As can be seen from the table, the outputs obtained in the proposed method are in agreement with the exact solution, and the larger the $m'$ value, the better the numerical approximation becomes. The exact and proposed method solutions for $\alpha = 1$, $\beta = 1$ are also plotted in Figure 5, whereas the solution graphs $u(t)$ and $v(t)$ for

the fractional orders $\alpha, \beta$ with integer orders are given in Figure 6. Again, the fractional orders are chosen to be not equal in the simulation. The numerical simulation is done for the fractional order pairs $\alpha = 0.25$, $\beta = 0.35$, $\alpha = 0.5$, $\beta = 0.6$, and $\alpha = 0.75$, $\beta = 0.85$, and the integer orders. As Figures 5 and 6 demonstrate, the solution approaches to the exact solution when $\alpha, \beta$ approach one, which thus validates the fractional solutions.

**Table 3.** Output values $u(t)$ and $v(t)$ of the proposed method for $\alpha = 1$, $\beta = 1$ and several $m'$ values with the exact solution of example 3.

| $t$ | Exact Solution | | $m'=24$ | | $m'=48$ | | $m'=96$ | | $m'=192$ | |
|---|---|---|---|---|---|---|---|---|---|---|
| | $u(t)$ | $v(t)$ | $u(t)$ | $v(t)$ | $u(t)$ | $v(t)$ | $u(t)$ | $v(t)$ | $u(t)$ | $v(t)$ |
| 0 | 1.0000 | 0.0000 | 1.0001 | 0.0005 | 1.0000 | 0.0001 | 1.0000 | 0.0000 | 1.0000 | 0.0000 |
| 0.1 | 1.0513 | 0.1105 | 1.0513 | 0.1111 | 1.0513 | 0.1107 | 1.0513 | 0.1106 | 1.0513 | 0.1105 |
| 0.2 | 1.1052 | 0.2443 | 1.1052 | 0.2450 | 1.1052 | 0.2444 | 1.1052 | 0.2443 | 1.1052 | 0.2443 |
| 0.3 | 1.1618 | 0.4050 | 1.1619 | 0.4058 | 1.1619 | 0.4052 | 1.1618 | 0.4050 | 1.1618 | 0.4050 |
| 0.4 | 1.2214 | 0.5967 | 1.2215 | 0.5977 | 1.2214 | 0.5970 | 1.2214 | 0.5968 | 1.2214 | 0.5967 |
| 0.5 | 1.2840 | 0.8244 | 1.2841 | 0.8257 | 1.2840 | 0.8247 | 1.2840 | 0.8244 | 1.2840 | 0.8244 |
| 0.6 | 1.3499 | 1.0933 | 1.3499 | 1.0947 | 1.3499 | 1.0936 | 1.3499 | 1.0934 | 1.3499 | 1.0933 |
| 0.7 | 1.4191 | 1.4096 | 1.4192 | 1.4114 | 1.4191 | 1.4101 | 1.4191 | 1.4097 | 1.4191 | 1.4097 |
| 0.8 | 1.4918 | 1.7804 | 1.4919 | 1.7825 | 1.4919 | 1.7810 | 1.4918 | 1.7806 | 1.4918 | 1.7805 |
| 0.9 | 1.5683 | 2.2136 | 1.5684 | 2.2161 | 1.5683 | 2.2143 | 1.5683 | 2.2138 | 1.5683 | 2.2137 |

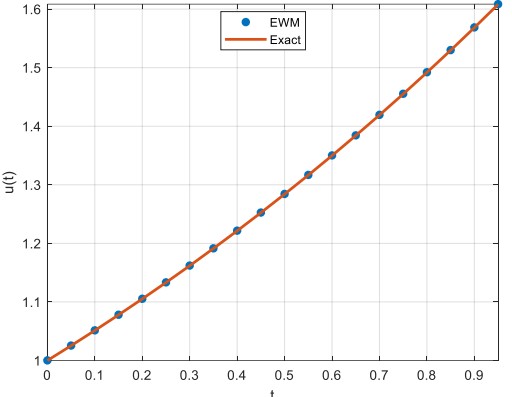 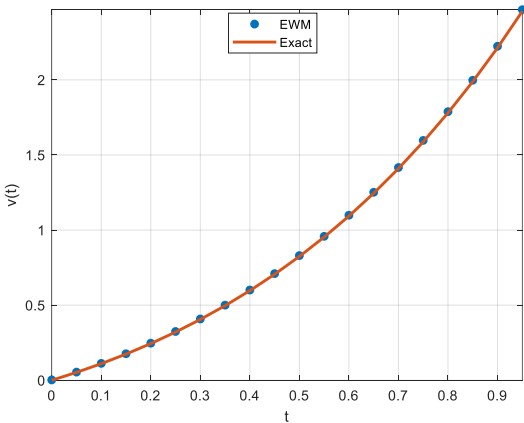

**Figure 5.** Numerical and exact solutions of $u(t)$ and $v(t)$ for $\alpha = 1$, $\beta = 1$ for Example 3 ($m' = 12$).

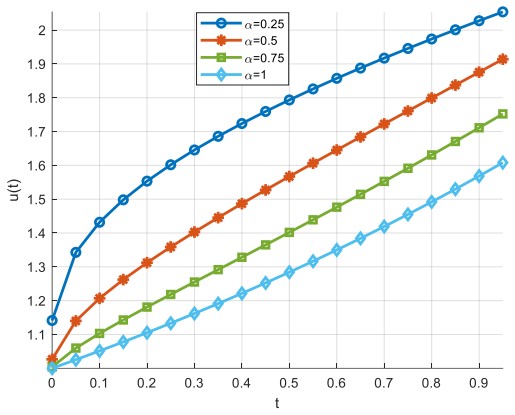 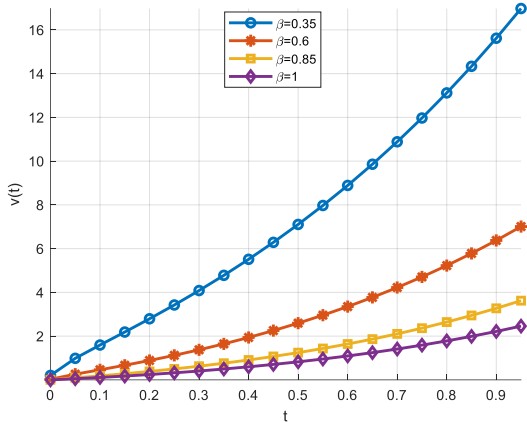

**Figure 6.** Numerical solutions of $u(t)$ and $v(t)$ for several fractional orders for Example 3 ($m' = 24$).

*4.4. Example 4*

For our last example, we consider the following system of FDEs

$$
\begin{aligned}
D^\alpha u(t) &= u(t) &, 0 < \alpha \le 1 \\
D^\beta v(t) &= 2u^2(t) &, 0 < \beta \le 1 \\
D^\gamma w(t) &= 3u(t)v(t) &, 0 < \gamma \le 1
\end{aligned}
\tag{45}
$$

with the initial conditions $u(0) = 1$, $v(0) = 1$, $w(0) = 1$ and the exact solution for $\alpha = 1$, $\beta = 1$ and $\gamma = 1$ is given as $u_{ex}(t) = e^t$, $v_{ex}(t) = e^{2t}$, and $w_{ex}(t) = e^{3t}$, $t \in [0,1]$.

Applying the proposed method to this system of FDEs, we have:

$$
\begin{aligned}
D^\alpha u(t) &\approx R_{m'}^T \, \psi(t) \\
D^\beta v(t) &\approx S_{m'}^T \, \psi(t) \\
D^\gamma w(t) &\approx T_{m'}^T \, \psi(t)
\end{aligned}
\tag{46}
$$

where $R_{m'}^T = [r_1, r_2, \ldots, r_{m'}]$, $S_{m'}^T = [s_1, s_2, \ldots, s_{m'}]$, and $T_{m'}^T = [t_1, t_2, \ldots, t_{m'}]$ are the unknown coefficients.

$$
u(t) = I^\alpha D^\alpha u(t) + u(0) \approx R_{m'}^T \, P_{m' \times m'}^\alpha \psi(t) + 1 \approx \underbrace{R_{m'}^T \, P_{m' \times m'}^\alpha \phi_{m' \times m'}}_{H_{m'}^T} B_{m'}(t) + 1
$$

$$
v(t) = I^\beta D^\beta v(t) + v(0) \approx S_{m'}^T \, P_{m' \times m'}^\beta \psi(t) + 1 \approx \underbrace{S_{m'}^T \, P_{m' \times m'}^\beta \phi_{m' \times m'}}_{K_{m'}^T} B_{m'}(t) + 1
\tag{47}
$$

$$
w(t) = I^\gamma D^\gamma w(t) + w(0) \approx T_{m'}^T \, P_{m' \times m'}^\gamma \psi(t) + 1 \approx \underbrace{T_{m'}^T \, P_{m' \times m'}^\gamma \phi_{m' \times m'}}_{L_{m'}^T} B_{m'}(t) + 1
$$

Substituting (46)–(47) in (45) we obtain the system of algebraic equations with $3m'$ unknowns in the form of $R_{m'}^T$, $S_{m'}^T$, and $T_{m'}^T$ coefficients:

$$
\begin{aligned}
R_{m'}^T \, \phi_{m' \times m'} &= H_{m'}^T + [1, 1, \ldots, 1]_{1 \times m'}, \\
S_{m'}^T \, \phi_{m' \times m'} &= 2\left[ \left(H_{m'}^T\right)^2 + 2H_{m'}^T + [1, 1, \ldots, 1]_{1 \times m'} \right], \\
T_{m'}^T \, \phi_{m' \times m'} &= 3\left(H_{m'}^T * K_{m'}^T\right) + 3H_{m'}^T + 3K_{m'}^T + [3, 3, \ldots, 3]_{1 \times m'}
\end{aligned}
\tag{48}
$$

The absolute errors obtained from the proposed method for $u(t)$, $v(t)$, and $w(t)$ for $m' = 48$ and $m' = 96$ ($\alpha = 1$, $\beta = 1$, and $\gamma = 1$) are summarized in Table 4. $E_u$, $E_v$, and $E_w$ represent the absolute errors in $u(t)$, $v(t)$, and $w(t)$, respectively. The absolute errors are roughly on the order of $10^{-3}$–$10^{-4}$ for $m' = 48$, and on the order of $10^{-4}$–$10^{-5}$ for $m' = 96$. As can be seen from the table, the absolute errors decrease with the larger $m'$ values, as expected. The numerical solutions and exact solutions of $u(t)$, $v(t)$, and $w(t)$ for the integer orders $\alpha = 1$, $\beta = 1$, and $\gamma = 1$ are plotted in Figure 7. Figure 7 shows that the numerical solution is very close to the exact solution even for the relatively small value of $m' = 12$. Also, the numerical solutions of $u(t)$, $v(t)$, and $w(t)$ for the fractional orders 0.75, 0.85, 0.95 with the integer orders one are plotted in Figure 8. As is the case with the previous examples, the solution approaches to the exact solution when $\alpha, \beta, \gamma$ approach one.

**Table 4.** Output values $u(t)$, $v(t)$, and $w(t)$ of the proposed method for $\alpha = 1$, $\beta = 1$, and $\gamma = 1$, and several $m'$ values with the exact solution of Example 4.

| $t$ | $m'=48$ | | | $m'=96$ | | |
|---|---|---|---|---|---|---|
| | $E_u$ | $E_v$ | $E_w$ | $E_u$ | $E_v$ | $E_w$ |
| 0 | $5.72 \times 10^{-5}$ | $2.41 \times 10^{-4}$ | $5.70 \times 10^{-4}$ | $1.39 \times 10^{-5}$ | $5.71 \times 10^{-5}$ | $1.32 \times 10^{-4}$ |
| 0.1 | $6.44 \times 10^{-5}$ | $2.78 \times 10^{-4}$ | $7.14 \times 10^{-4}$ | $1.60 \times 10^{-5}$ | $6.82 \times 10^{-5}$ | $1.74 \times 10^{-4}$ |

**Table 4.** *Cont.*

| $t$ | $m'$=48 | | | $m'$=96 | | |
|---|---|---|---|---|---|---|
| | $E_u$ | $E_v$ | $E_w$ | $E_u$ | $E_v$ | $E_w$ |
| 0.2 | $7.48 \times 10^{-5}$ | $3.42 \times 10^{-4}$ | $9.76 \times 10^{-4}$ | $1.87 \times 10^{-5}$ | $8.57 \times 10^{-5}$ | $2.45 \times 10^{-4}$ |
| 0.3 | $8.82 \times 10^{-5}$ | $4.39 \times 10^{-4}$ | $1.40 \times 10^{-3}$ | $2.20 \times 10^{-5}$ | $1.09 \times 10^{-4}$ | $3.50 \times 10^{-4}$ |
| 0.4 | $1.02 \times 10^{-4}$ | $5.40 \times 10^{-4}$ | $1.90 \times 10^{-3}$ | $2.57 \times 10^{-5}$ | $1.37 \times 10^{-4}$ | $4.86 \times 10^{-4}$ |
| 0.5 | $1.24 \times 10^{-4}$ | $7.53 \times 10^{-4}$ | $3.05 \times 10^{-3}$ | $3.04 \times 10^{-5}$ | $1.80 \times 10^{-4}$ | $7.15 \times 10^{-4}$ |
| 0.6 | $1.39 \times 10^{-4}$ | $8.76 \times 10^{-4}$ | $3.78 \times 10^{-3}$ | $3.45 \times 10^{-5}$ | $2.15 \times 10^{-4}$ | $9.23 \times 10^{-4}$ |
| 0.7 | $1.60 \times 10^{-4}$ | $1.08 \times 10^{-3}$ | $5.07 \times 10^{-3}$ | $4.00 \times 10^{-5}$ | $2.70 \times 10^{-4}$ | $1.27 \times 10^{-3}$ |
| 0.8 | $1.86 \times 10^{-4}$ | $1.37 \times 10^{-3}$ | $7.14 \times 10^{-3}$ | $4.64 \times 10^{-5}$ | $3.42 \times 10^{-4}$ | $1.78 \times 10^{-3}$ |
| 0.9 | $2.13 \times 10^{-4}$ | $1.69 \times 10^{-3}$ | $9.56 \times 10^{-3}$ | $5.35 \times 10^{-5}$ | $4.28 \times 10^{-4}$ | $2.44 \times 10^{-3}$ |

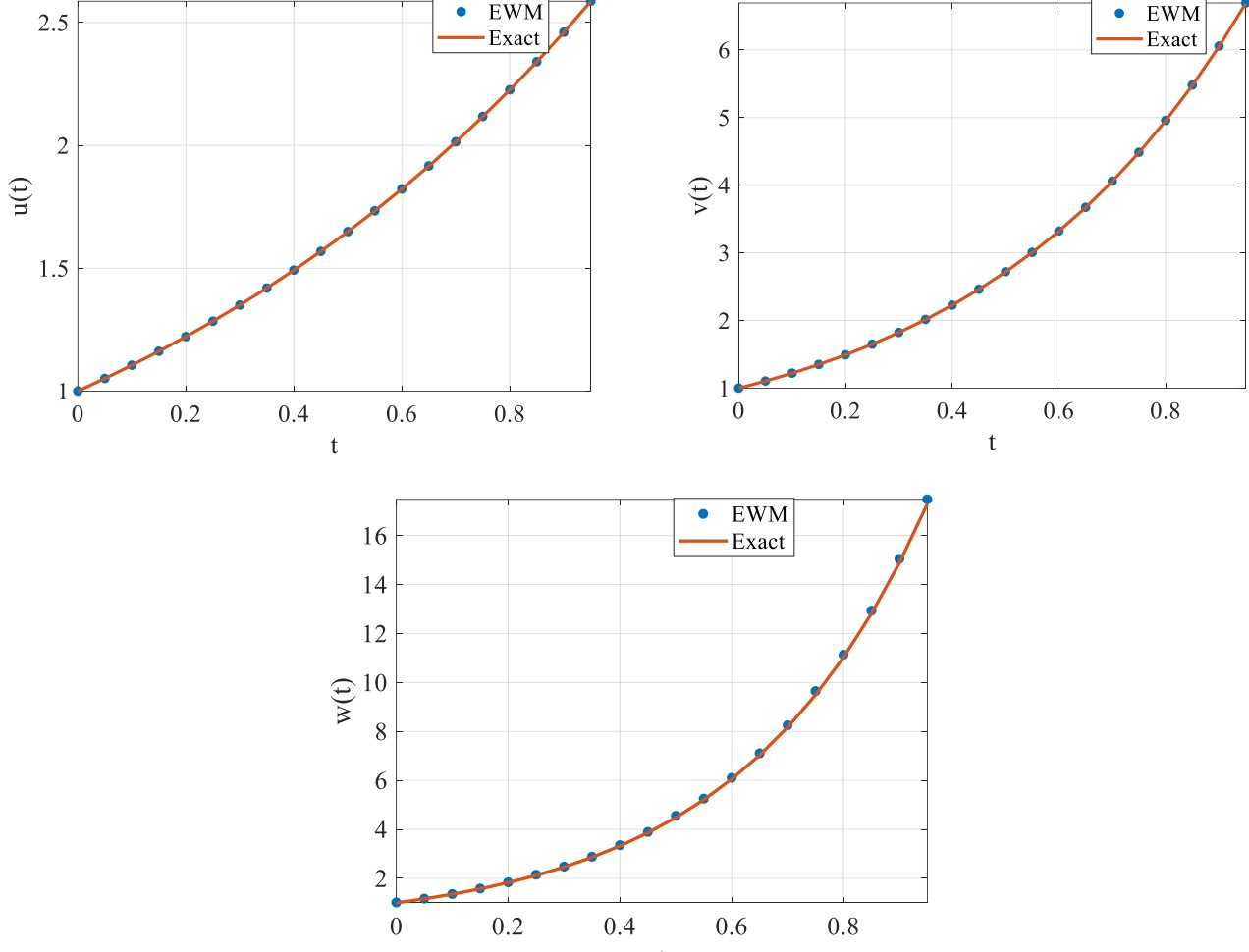

**Figure 7.** Numerical and exact solutions for $u(t)$, $v(t)$ and $w(t)$ for $\alpha = 1$, $\beta = 1$, and $\gamma = 1$ for Example 4 ($m' = 12$).

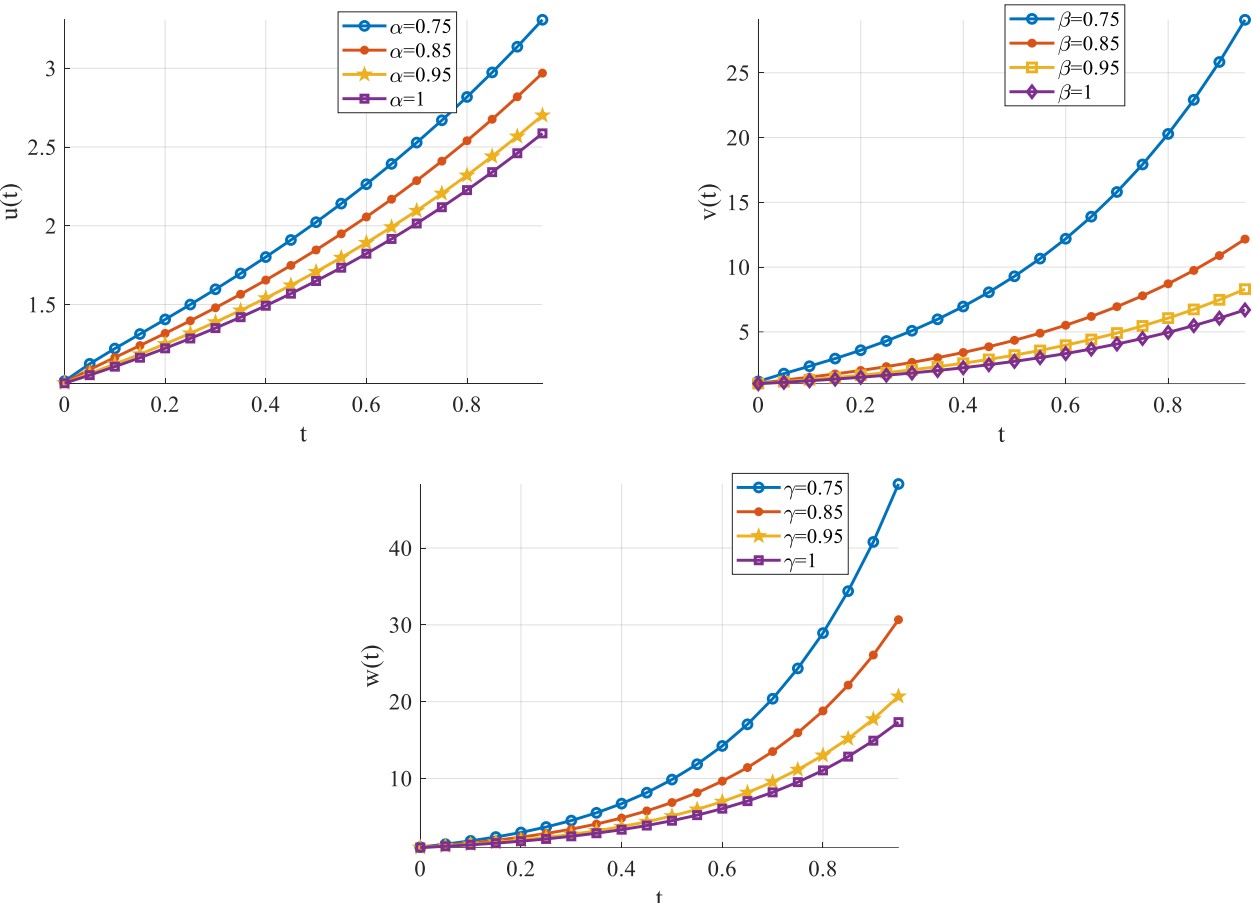

**Figure 8.** Numerical solutions of $u(t)$, $v(t)$, and $w(t)$ for several fractional orders for Example 4 ($m' = 24$).

The above four examples demonstrate that the proposed algorithm is effective and accurate to apply to the systems of FDEs.

## 5. Conclusions

In this paper, we proposed a numerical solution method for the systems of FDEs. The proposed method employs discrete Euler wavelets to obtain the so-called operational matrices for the fractional integration. With the help of the block pulse functions and the small number of terms in the Euler polynomials, the operational matrices become very sparse. This is the essence of the proposed method since sparse operational matrices used for the mapping between fractional terms in the FDEs and the discrete terms in the numerical approach determine the required memory and also determine the speed of the numerical method. The system of FDEs is converted into a system of algebraic equations and the algebraic equation system is solved using the Newton–Raphson method to obtain the unknown coefficients. The accuracy of the proposed method increases for the larger number of collocation points, as expected. This is verified in each of the considered examples shown in Tables 1–4 and Figures 1, 3, 5 and 7. The maximum errors are generally on the order of $10^{-4}$–$10^{-6}$ for the collocation points up to $m' = 96$. For higher accuracy, the number of the collocation points must be increased. The numerical solutions obtained for fractional orders, which are the main target of this study, are also in agreement with the solutions obtained for integer orders. One can see from Figures 2, 4, 6 and 8 that the method provides precise solutions for the fractional orders owing to the fact that when the fractional orders approach the integer values, the solution also approaches the exact solution obtained for the integer orders.

For future work, the proposed method can also be applied to the systems of variable-order FDEs, systems of fractional partial differential equations, systems of fractional integral equations, and systems of fractional delay differential equations.

**Author Contributions:** Methodology, A.T.D.; Software, S.N.T.P.; Formal analysis, A.T.D. All authors have read and agreed to the published version of the manuscript.

**Funding:** This work has been supported by Yildiz Technical University Scientific Research Projects Coordination Unit under project number FBA-2023-5416.

**Institutional Review Board Statement:** Not applicable.

**Informed Consent Statement:** Not applicable.

**Data Availability Statement:** Not applicable.

**Conflicts of Interest:** The authors declare no conflict of interest.

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
