# Peer review of "Euler Wavelet Method as a Numerical Approach for the Solution of Nonlinear Systems of Fractional Differential Equations"

_fractalfract, doi:10.3390/fractalfract7030246_

Round 1

Reviewer 1 Report

The main purpose of this study is a numerical approach for solving systems of nonlinear fractional differential equations.  The method's major objective is to transform the nonlinear FDE into a nonlinear system of algebraic equations that is straightforward to solve with matrix techniques.These new types' which weren't realized before by any other method have been detected via four different techniques. The referee focus on (1) how about the background of eqs.(33) and (38), the physical and optical problems described by the equations should be given. (2) on solving nonlinear pdes, some useful method have been employed and some recent results are useful to be cited. (3) the conclusion is so simple, which should be extened in some sense.

Author Response

  • how about the background of eqs.(33) and (38), the physical and optical problems described by the equations should be given.

Answer: Instead of focusing on the particular examples systems of FDEs, the general application areas of the systems of FDEs are included in the introduction section.

  • on solving nonlinear pdes, some useful method have been employed and some recent results are useful to be cited.

Answer: The introduction section is extended by citing the methods used in the systems of PDEs as well.

(3) the conclusion is so simple, which should be extened in some sense.

Answer: The conclusion section is extended as requested.

Reviewer 2 Report

The paper deals with an important task. It has a scientific novelty. It has a logical structure. The proposed approach is logical. The introduction and experimental sections must be improved.

Suggestions: 

1.     Review of literary sources is done superficially. For example, the authors indicate: "a variety of numerical methods have been developed by many researchers recently". It is appropriate to analyze them, group them, highlight their advantages, disadvantages, and possible directions of research development.

2.     Introduction section should be extended using more clearly the motivation for this paper.

3.     To validate the proposed approach, the authors use the solutions obtained for a non-fractal problem, that is, using a traditional differentiation operator. And this, in turn, is the Cauchy problem, for finding which there are well-known and proven methods, such as the Runge-Kutta method. Therefore, I consider validating the method using the fractional derivative in the traditional case impractical.

4.     The results of using the proposed approach are shown in graphs and tables, but there is no analysis of them. The advantages of using the proposed approach are not emphasized.

5.     It would be appropriate to expand the overview section with a review of methods for finding the solution for the mathematical models based on the use of fractional order derivatives, such as approximation using splines, using the finite element method and others. For example, pay attention to publications DOI: 10.3390/a15020069, DOI: 10.3390/e22111328 and DOI: 10.1109/CSIT49958.2020.9321996

6.     Conclusion section should be extended using: 

- numerical results obtained in the paper;

- limitations of the proposed approach;

          - prospects for future research.

Author Response

  1. Review of literary sources is done superficially. For example, the authors indicate: "a variety of numerical methods have been developed by many researchers recently". It is appropriate to analyze them, group them, highlight their advantages, disadvantages, and possible directions of research development.

Answer: The introduction section is extended as requested.

  1. Introduction section should be extended using more clearly the motivation for this paper.

Answer: The introduction section is extended as requested.

  1. To validate the proposed approach, the authors use the solutions obtained for a non-fractal problem, that is, using a traditional differentiation operator. And this, in turn, is the Cauchy problem, for finding which there are well-known and proven methods, such as the Runge-Kutta method. Therefore, I consider validating the method using the fractional derivative in the traditional case impractical.

Answer: As the reviewer suggests, the solutions for nonfractional-orders are superficial, there are a lot of well developed numerical methods for the ODEs and PDEs. However, they are included in the manuscript to compare the solutions obtained for fractional orders. The validity of the solutions for the fractional orders are generally proved by the convergence analyses in wavelet methods. The convergence analysis for Euler wavelet operational matrix method is given by the ref [35] as we cited in the original version. Also, the numerical solution obtained for the fractional orders should converge to the exact solution when the orders approximate to the integer values. This is a good indicator.

  1. The results of using the proposed approach are shown in graphs and tables, but there is no analysis of them. The advantages of using the proposed approach are not emphasized.

Answer: Brief analyses and comparisons had already been included where each of the tables and figures are cited in the manuscript. Those regions have been extended in the revised version as suggested.

  1. It would be appropriate to expand the overview section with a review of methods for finding the solution for the mathematical models based on the use of fractional order derivatives, such as approximation using splines, using the finite element method and others. For example, pay attention to publications DOI: 10.3390/a15020069, DOI: 10.3390/e22111328 and DOI: 10.1109/CSIT49958.2020.9321996

Answer: The introduction section is extended including some of the requested references. Spline methods had already been included in [16-17] in the original manuscript. The finite element method paper is added as ref [32].

  1. Conclusion section should be extended using: 

- numerical results obtained in the paper;

- limitations of the proposed approach;

          - prospects for future research.

Answer: The conclusion section is extended as requested. To the best of our knowledge, we have not encountered any limitations, on the contrary, the method is fast and accurate with low computational load. Numerical results and prospects for future research are added.

Reviewer 3 Report

In the manuscript fractalfract-2243596, the authors present a numerical approach for solving systems of nonlinear fractional differential equations. In addition to the description of the method, the paper also contains numerical results obtained using the method described by the authors.

The topic is interesting, but a revision of this article is required before any decision can be made.
I have included some questions below.

1. In my opinion, in the conclusion, the authors should describe whether the proposed method has any limitations.

2. In examples 1-3, in appropriate tables, the authors presented the values of absolute errors. Why didn't the authors attach such a table to Example 4?

3. In Examples 2-4, in Fig. Figures 2, 4, and 6, the authors showed a comparison of numerical and exact solutions. Why didn't the authors add a similar figure to Example 1?

4. In Figs. 1 and 7, the authors showed the results for u(t), v(t), and w(t) for the same values of alpha and beta. Why did the authors plot u(t) results for alpha =0.4, 0.6, 0.8, 1, and v(t) results for betha = 0.5,0.7, 0.9, 1 in fig. 3? Similar situation we have in Fig. 5. Is it intentional? If so, could the authors explain it?  

5. In all Figures, descriptions of axis and legends are not well visible. The size of the font should be increased.

Author Response

  1. In my opinion, in the conclusion, the authors should describe whether the proposed method has any limitations.

Answer: The conclusion section is extended as requested.

  1. In examples 1-3, in appropriate tables, the authors presented the values of absolute errors. Why didn't the authors attach such a table to Example 4?

Answer: Since the solution has 3 outputs instead of 2 for other examples, such a table would take more space, therefore we did not include the table results for the example 4 to avoid clutter. We include the table for example4 in the revised version.

  1. In Examples 2-4, in Fig. Figures 2, 4, and 6, the authors showed a comparison of numerical and exact solutions. Why didn't the authors add a similar figure to Example 1?

Answer: Figure 1 is added as requested, therefore all the names of the figures in the revised version is increased by 1 compared to the original version.

  1. In Figs. 1 and 7, the authors showed the results for u(t), v(t), and w(t) for the same values of alpha and beta. Why did the authors plot u(t) results for alpha =0.4, 0.6, 0.8, 1, and v(t) results for betha = 0.5,0.7, 0.9, 1 in fig. 3? Similar situation we have in Fig. 5. Is it intentional? If so, could the authors explain it?  

Answer: The fractional orders can be chosen arbitrarily, hence can be chosen different form one another. We have chosen several different fractional orders to demonstrate the method is powerful and accurate.

5. In all Figures, descriptions of axis and legends are not well visible. The size of the font should be increased.

Answer: All the figures are redrawn to have larger legends and values to improve visibility.

Round 2

Reviewer 2 Report

The authors took into account the recommendations.

Reviewer 3 Report

The authors have modified the manuscript.
Therefore, I recommend the publication of the manuscript.